# Genome characterization based on the Spike-614 and NS8-84 loci of SARS-CoV-2 reveals two major possible onsets of the COVID-19 pandemic

Xiaowen Hu[1,2]☯, Yaojia Mu[1]☯, Ruru Deng[1]☯, Guohui Yi[3], Lei Yao[4]*, Jiaming Zhang●[1]*

**1** Key Laboratory of Microbiology of Hainan Province, Institute of Tropical Bioscience and Biotechnology, Chinese Academy of Tropical Agricultural Sciences, Haikou, Hainan, China, **2** Institute of South Subtropical Crops, Chinese Academy of Tropical Agricultural Sciences, Zhanjiang, Guangdong, China, **3** Public Research Laboratory, Hainan Medical University, Haikou, Hainan, China, **4** Sichuan Provincial Key Laboratory for Human Disease Gene Study and the Center for Medical Genetics, Department of Laboratory Medicine, Sichuan Academy of Medical Sciences & Sichuan Provincial People's Hospital, University of Electronic Science and Technology, Chengdu, China

☯ These authors contributed equally to this work.
* zhangjiaming@itbb.org.cn (JZ); yaolei2009@gmail.com (LY)

**Data Availability Statement:** All relevant data are within the paper and its Supporting Information files.

## Abstract

The global COVID-19 pandemic has lasted for 3 years since its outbreak, however its origin is still unknown. Here, we analyzed the genotypes of 3.14 million SARS-CoV-2 genomes based on the amino acid 614 of the Spike (S) and the amino acid 84 of NS8 (nonstructural protein 8), and identified 16 linkage haplotypes. The GL haplotype (S_614G and NS8_84L) was the major haplotype driving the global pandemic and accounted for 99.2% of the sequenced genomes, while the DL haplotype (S_614D and NS8_84L) caused the pandemic in China in the spring of 2020 and accounted for approximately 60% of the genomes in China and 0.45% of the global genomes. The GS (S_614G and NS8_84S), DS (S_614D and NS8_84S), and NS (S_614N and NS8_84S) haplotypes accounted for 0.26%, 0.06%, and 0.0067% of the genomes, respectively. The main evolutionary trajectory of SARS-CoV-2 is DS→DL→GL, whereas the other haplotypes are minor byproducts in the evolution. Surprisingly, the newest haplotype GL had the oldest time of most recent common ancestor (tMRCA), which was May 1 2019 by mean, while the oldest haplotype DS had the newest tMRCA with a mean of October 17, indicating that the ancestral strains that gave birth to GL had been extinct and replaced by the more adapted newcomer at the place of its origin, just like the sequential rise and fall of the delta and omicron variants. However, the haplotype DL arrived and evolved into toxic strains and ignited a pandemic in China where the GL strains had not arrived in by the end of 2019. The GL strains had spread all over the world before they were discovered, and ignited the global pandemic, which had not been noticed until the virus was declared in China. However, the GL haplotype had little influence in China during the early phase of the pandemic due to its late arrival as well as the strict transmission controls in China. Therefore, we propose two major onsets of the COVID-19 pandemic, one was mainly driven by the haplotype DL in China, the other was driven by the haplotype GL globally.

**Funding:** This research was supported by grants from National Key R&D Program of China and the Central Public-interest Scientific Institution Basal Research Fund to J.Z. (1630052020022), and the Project of Science and Technology Department of Sichuan Provincial of China to L.Y. (2019JDJQ0035). The funders had no role in study design, data collection and analysis, decision to publish, or preparation of the manuscript.

**Competing interests:** The authors have declared that no competing interests exist.

## Introduction

The severe acute respiratory syndrome coronavirus 2 (SARS-CoV-2) is the causal agent of COVID-19, a disease first reported in Wuhan, China [1–3]. This virus differs from all known coronaviruses, including the six coronavirus species that are known to cause human disease [4], and has been classified as the seventh coronavirus that can infect humans by the International Committee on Taxonomy of Viruses [5]. SARS-CoV-2 quickly spread within China. A range of public health interventions was used to control the epidemic, including isolation of suspected and confirmed cases, transport prohibition in and out of epidemic centers, suspension of public transport, closing of schools and entertainment venues, and public gathering bans and health checks [6, 7]. These measures eventually brought about an end to sustained local transmission across China in April 2020. However, COVID-19 expands its territories, and became a global pandemic. As of 23 April 2023, there have been over 763 million confirmed cases and 6.9 million deaths of COVID-19, reported to WHO (https://covid19.who.int/).

To our knowledge, the origins of SARS-CoV-2 remain elusive. Understanding how, when, and where the virus was transmitted from its natural reservoir to humans is crucial for preventing future coronavirus outbreaks [8]. Progresses have been made in this aspect. A bat-origin virus RaTG13 that has 96% sequence identities with SARS-CoV-2 has been published [9], which suggests bats as a likely natural reservoir of this virus. Pangolin [10], snakes [11], turtles [12], and/or Bovidae and Cricetidae [13] have been suggested to act as potential intermediate hosts that helped the virus to cross the species barriers to infect humans. According to the clinical summary of the earliest cases of COVID-19 (also known as 2019-nCoV) reported in China, the majority of cases were exposed to the Huanan Seafood Market [14], which also had wild animals. Therefore, the market was considered to be an obvious candidate location of the initial zoonotic transmission event of COVID-19 [15], and both epidemiological and phylogenetic approaches suggested late 2019 to be the occurring time [14, 16, 17]. However, none of the wild animals from the Huanan Seafood Market were tested positive for SARS-CoV-2 [18]. Some environmental samples were positive, but their viral genomes were not located at the basal position of the phylogeny [19]. Moreover, some of the early cases were not epidemiologically linked to the Huanan Seafood Market [16], but linked to other markets [20], and animal-to-human transmission in the market has never been confirmed and should not be overemphasized [21].

The origin and evolution of SARS-CoV-2 has been well reviewed [22, 23]. Most tracing research focused on analysis of the early genomes [23–25]. For example, Tang and colleagues classified the early SARS-CoV-2 genomes into two major lineages (L and S lineages), and the two lineages were well defined by just two SNPs that show complete linkage across SARS-CoV-2 strains [26]. The L lineage constituted 70% of the sequenced genomes, while the S lineage constituted approximately 30% of the sequenced genome in the 103 genomes in the early phase of epidemic [26]. However, higher percentage is not sufficient to substantiate a more aggressive type [27], until the emergence of the delta variant [28]. Evolutionary analyses suggested that the S lineage appeared to be more related to coronaviruses in animals [26]. Li and colleagues outlined the early viral spread in Wuhan and its transmission patterns in China and across the rest of the world [23]. Pekar and colleagues analyzed the early viral genomes (before the end of April 2020) in China, and defined a period between mid-October and mid-November 2019 as the plausible interval when the first case of SARS-CoV-2 emerged in Hubei province, China [24]. Ruan and colleagues discovered distinct set of mutations driving the waves of replacements of strains, and the split between the Asian and European lines occurred before September of 2019, suggesting twin-beginning scenario of the pandemic [25]. Due to the presumed undetected transmission and insufficient genome sequences in the early phase of the pandemic, the origin of the virus remains under cover. It is true that coronaviruses evolve

rapidly [29], and the mutation rate of SARS-CoV-2 was predicted to be approximately $8.0 \times 10^{-4}$ subs/site/year [30]. However, the inheritance rate of each nucleotide should be higher than 99.9992%, taken the negative selection pressure into account [26]. Upon the increased coverage of the sequenced genomes in the late phase of the pandemic, we believe that the secret of the viral origin may be buried in the genomes, not necessarily in the early genomes.

In this study, we analyzed 3.14 million genomes obtained from GISAID database, and used two gene loci to anchor the haplotypes, and estimated the time of most recent common ancestor (tMRCA) and the proximal geographical origin of SARS-CoV-2. The evolutionary trajectory of the SARS-CoV-2 haplotypes is proposed.

## Materials and methods

The findings of this study are based on metadata associated with 3,244,841 SARS-CoV-2 genome sequences available on GISAID up to 2021-10-18, via gisaid.org/ EPI_SET_230423ot (S1 File). The genomes that were incomplete (<29,000 nt), had low coverage with >0.5% unique amino acid mutations and more than 1% 'N's were filtered. A total of 3,140,626 sequences were remained for further analysis. All the genomes were isolated from human patients. To analyze the temporal-dependent evolutionary pattern of SARS-CoV-2, the genome sequences were divided into groups by months according to their sample collection date, and were aligned using ViralMSA [31] and Clustal-Omega [32]. The unknown and/or low-quality nucleotides were removed from the alignments using personalized scripts, which is publicly available at github (https://github.com/XiaowenH/SeqSC.git). Nucleotide diversity (Pi) and population mutation rate (Theta-W) of each subgroup was calculated using Pairwise Deletion Model with DnaSP 6 [33]. The Pearson correlation coefficient [34] and significant analysis was calculated by cor.test in the R package (version 4.1.1) developed by the R Development Core Team (https://www.r-project.org).

To genotype the genomes based on the spike gene, the spike coding sequences of the 3.14 million genomes were aligned to the reference genome Wuhan-Hu-1 (MN908947.3), and the relevant CDS was extracted and translated into peptides by using SeqKit package [35]. The peptides were aligned with the reference using Clustal-Omega [32], and the sites aligned with the site 614 of the reference was identified using personalized script (https://github.com/XiaowenH/SeqSC.git). The NS8 gene was genotyped using the same method, except that NS8 gene was used as the reference.

To analyze the mutation accumulation pattern of each haplotype, the genome sequences were divided by month, and were aligned using ViralMSA [31] and/or Clustal-Omega [32]. Phylogenetic analysis was performed using Mega 11 [36]. The evolutionary network analysis was performed by using the Median-joining Networks in Potpart 1.7 [37].

To explore the evolutionary dynamics, genomes of each haplotype were aligned with ViralMSA [31] after removal of redundant sequences by using CD-HIT [38] with a threshhold of 99.98%. The outputs were transformed to Nexus format by using ALTER [39]. Bayesian phylodynamics analysis was performed by using BEAST 1.10 [40] with molecular clock set to strict and coalescent prior set to Bayesian Skyline [24]. The posterior distribution was summarized by using TRACER 1.7 [41].

## Results

### Mutation accumulation dynamics of SARS-CoV-2

A total of 3,140,626 SARS-CoV-2 genomes designated as complete and high coverage were obtained in the GISAID database (www.gisaid.org) on 2021-10-18. The average length of the genomes is 29,781.5 bases with the minimum and maximum lengths of 29,000 and 31,579

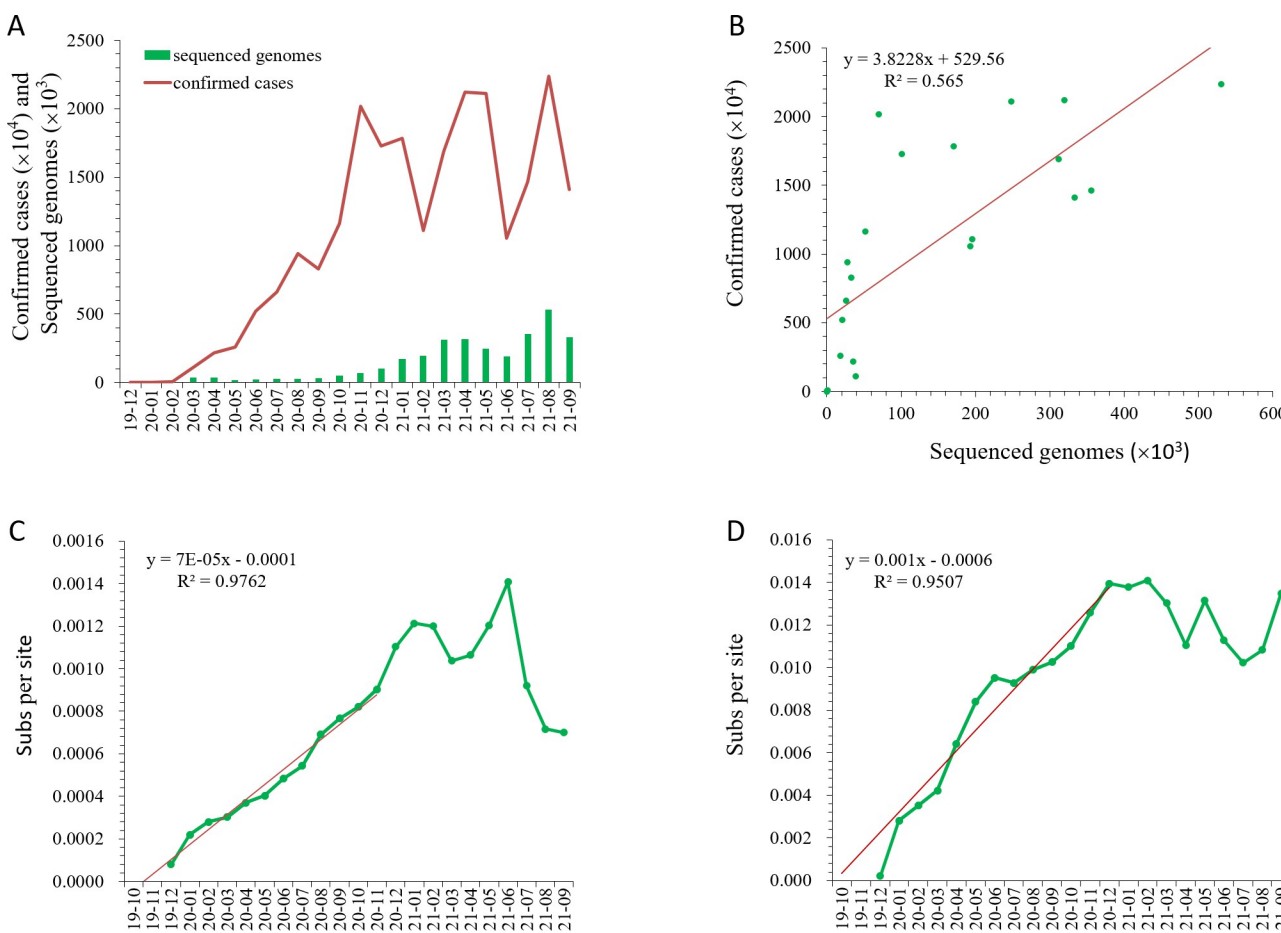

**Fig 1. Accumulation dynamics of sequenced genomes, confirmed cases and mutations of SARS-CoV-2.** A, Accumulation of sequenced genome and confirmed cases. B, Correlation of sequenced genomes and confirmed cases. C, Accumulation dynamics of nucleotide diversity (Pi) in genomes as calculated using pairwise deletion (PD) model; D, Accumulation dynamics of theta (W).

bases, respectively. Both the sequenced genomes and confirmed cases increased exponentially during 2020, and remained at high level during 2021 (Fig 1A). Regression analysis shows that the sequenced genomes and confirmed cases are positively correlated with a linear coefficient r = 0.75 (p = 5.5e-05, Fig 1B), indicating that the sequenced genomes sufficiently represented the dynamic population.

The nucleotide diversity (Pi) was accumulated in a temporal-dependent pattern (Fig 1C, S1 Table). Both Pi and Theta-W (population mutation rate) increased linearly in 2020 with correlation coefficients r = 0.99 (p = 5.349e-10, Fig 1C) and r = 0.975 (p = 7.331e-08, Fig 1D) for Pi and Theta-W, respectively. They then varied up and down in 2021, possibly due to the rise and fall of dominant viral strains, such as the delta and the omicron variants. The rising of the delta and the omicron variants greatly decreased nucleotide diversity of viral population, and application of vaccines starting from 2021 may also have a selection pressure that decreased the diversity (unpublished data).

## Mutations of the amino acid 84 of NS8 in the 3.14 million genomes

The mutation at site 84 of the nonstructural protein ORF8 (NS8) was used to classified the early SARS-CoV-2 strains, and two major lineages (designated L and S) were well defined by

**Table 1. Alleles of the amino acid 84 in the nonstructural protein 8 (NS8).**

| Allele | Sequence (75–93) | Genomes | Percentage | First collect | First collection place |
|---|---|---|---|---|---|
| NS8_TG13 | DIGNYTVSC**S**PFTINCQEP | RaTG13 ref. | | 2013-07-24 | Yunan,China |
| NS8_wiv04 | DIGNYTVSC**L**PFTINCQEP | GISAID ref. | | 2019-12-30 | Wuhan,China |
| NS8_84L | DIGNYTVSC**L**PFTINCQEP | 3,115,900 | 99.6564* | 2019-12-24 | Wuhan,China |
| NS8_84S | DIGNYTVSC**S**PFTINCQEP | 10,732 | 0.3432* | 2019-12-30 | Wuhan,China |
| NS8_84V | DIGNYTVSC**V**PFTINCQEP | 3 | 0.0001* | 2020-09-15 | South Korea |
| NS8_84I | DIGNYTVSC**I**PFTINCQEP | 4 | 0.0001* | 2021-05-05 | USA |
| NS8_84F | DIGNYTVSC**F**PFTINCQEP | 2 | 0.00006* | 2020-02-02 | Sichuan,China |
| NS8_84C | DIGNYTVSC**C**PFTSNCQEP | 1 | 0.00003* | 2020-03-12 | USA |
| unknown | unknown | 13,985 | 0.445293 | 2020-01-14 | Japan |
| Total | | 3,140,626 | | | |

*Note: Percentage in the genomes that have been successfully classified.

two different SNPs (T8782C and C28144T, the latter being Orf8-S84L) [26]. The NS8 contains 121 amino acids, and promotes the expression of the ER unfolded protein response factor ATF6 in human cells [42]. We extracted the NS8 coding DNA sequence from all the 3.14 million SARS-CoV-2 genomes, and classified the genomes according to the identities at the amino acid 84. Besides the two known alleles, four additional mutations were identified, including NS8_84V, NS8_84I, NS8_84F, NS8_84C (Table 1). NS8_84L was the first to have been collected. Its first collection date was 2019-12-24 (EPI_ISL_402123), and a total of 21 NS8_84L samples were collected in Wuhan, China in December 2019, and constituted 95.5% of the early samples (S2 Table). This genotype constituted 99.66% of the 3.13 million genomes that were successfully classified (Table 1). NS8_84S was first collected on 2019-12-30 and constituted approximately 0.34% of the total genomes. The numbers of genomes with the newly identified alleles were very few (Table 1). Approximately 0.4% genomes were not classified due to incomplete sequencing or low quality in the region.

## Mutation accumulation and phylogenetic analysis of NS8 alleles

NS8_84L was dominant over NS8_84S all the time (S2 Table). The genome number of NS8_84L increased exponentially from December 2019 to March 2021, and kept at high numbers (Fig 2A), while NS8_84S had a peak in March 2020, and remained at low numbers the rest of time (Fig 2B). Even in its peak month, its genome number was only 12% of NS8_84L (S2 Table). The genome number of NS8_84L is positively correlated with the number of the confirmed cases with a coefficient r = 0.88 (p = 8.914e-05, Fig 2C), while the genome number of NS8_84S is not (r = -0.32, p = 0.29, Fig 2D). Therefore, NS8_84L was the main genotype driving the growth of confirmed cases worldwide.

The nucleotide diversity of both genotypes was accumulated linearly in 2020 (Fig 2E and 2F). Regression analysis revealed first occurrence dates of approximately 14 October 2019 (95% confidence interval: 18 September to 9 November) for the genotype NS8_84L, and 28 September 2019 (95% confidence interval: 6 August to 21 November) for genotype and NS8_84S, respectively. Therefore, NS8_84S occurred earlier than NS8_84L. Phylogenetic analysis using both peptide and RNA sequences of the NS8 showed that NS8_84S was closest to the root (Fig 2G and 2H), and the phylogenies by using different methods all suggested an earlier occurrence of NS8_84S than NS8_84L (S2 and S3 Figs), which was supported by the fact that the bat SARS-CoV viruses all have NS8_84S [26].

Taken together, NS8_84S is an earlier genotype than NS8_84L, and the latter may be originated from mutation of NS8_84S.

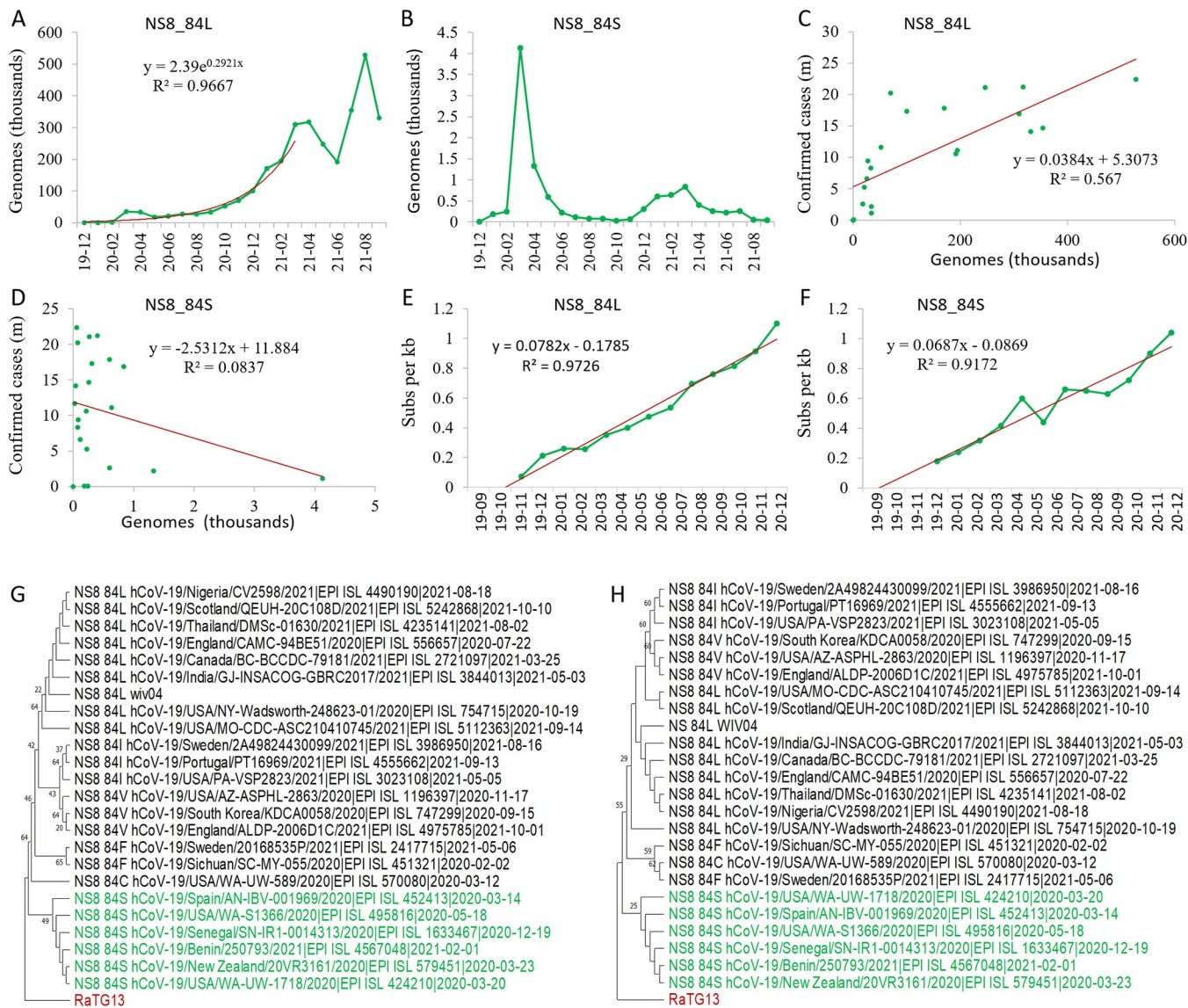

**Fig 2. Mutation accumulation and phylogenetic analysis of NS8 variants in SARS-CoV-2.** A and B, Genome accumulation curve of NS8_84L and NS8_84S as counted by the sample collection date; C and D, Correlation analysis between genomes of genotypes and confirmed cases; E and F, Mutation (Pi) accumulation dynamics of genotype NS8_84L (E) and NS8_84S (F); G, Maximum Likelihood tree of peptide sequences of NS8 as inferred by using the JTT matrix-based model [43]. The bootstrap supports by 1000 replicates are shown next to the branches. H, Maximum Likelihood tree of gene sequences of NS8 as inferred by using the Tamura-Nei model [44]. The bootstrap supports are shown above the branches. All positions with less than 95% site coverage were eliminated. Evolutionary analyses were conducted in MEGA11 [36].

## Mutations of the amino acid 614 of the spike protein in the 3.14 million genomes

Mutation of D614G in Spike (S) protein was the major driving force in the COVID-19 pandemic, the virulent strains alpha, delta, and omicron all carried this mutation. The S is a surface projection glycoprotein. Mutations in this protein have previously been associated with altered pathogenesis and virulence in other coronaviruses [45]. We extracted the S gene sequences from the 3.14 genomes, and classified the genomes based on mutations at the amino acid 614. Seven alleles were identified, including S_614D, S_614G, S_614N, S_614S, S_614V, S_614A, S_614C (Table 2).

**Table 2. Alleles of the amino acid 614 in the spike protein (spike_614) of SARS-CoV-2.**

| Allele | Sequence (605–623) | Genomes | Percentage | First collect | Collection place |
|---|---|---|---|---|---|
| S_TG13 | SNQVAVLYQ**D**VNCTEVPVA | reference | | 2013-07-24 | Yunnan, China |
| S_wiv04 | SNQVAVLYQ**D**VNCTEVPVA | reference | | 2019-12-30 | Wuhan, China |
| S_614G | SNQVAVLYQ**G**VNCTEVPVA | 2,846,092 | 99.293 | 2020-01-01 | France, Argentina |
| S_614D | SNQVAVLYQ**D**VNCTEVPVA | 19,992 | 0.697 | 2019-12-24 | Wuhan, China |
| S_614N | SNQVAVLYQ**N**VNCTEVPVA | 200 | 0.00698 | 2020-03-24 | England |
| S_614S | SNQVAVLYQ**S**VNCTEVPVA | 61 | 0.00213 | 2020-05-21 | USA |
| S_614A | SNQVAVLYQ**A**VNCTEVPVA | 11 | 0.00038 | 2020-07-13 | South Korea |
| S_614V | SNQVAVLYQ**V**VNCTEVPVA | 2 | 0.00007 | 2020-07-10 | USA |
| S_614C | SNQVAVLYQ**C**VNCTEVPVA | 1 | 0.00003 | 2021-07-30 | Cambodia |
| Total | | 3,140,626 | | | |

The S_614D was the earliest to have been collected, and was first collected in Wuhan, China on 2019-12-24 (EPI_ISL_402123, Table 2). In outside China, it was first collected in Thailand on 2020-01-08, Nepal on 2020-01-13, USA on 2020-01-19, and France on 2020-01-23. This genotype constituted the second largest number of the genomes with a percentage of 0.7%.

The S_614G accounted for 99.69% of the genomes (Table 2), and was first collected on 2020-01-01 in Argentina (EPI_ISL_4405694), followed by nine samples collected on 2020-01-03 in USA (EPI_ISL_3537067, EPI_ISL_3537066, EPI_ISL_3537065, EPI_ISL_3537064, EPI_ISL_3537063, EPI_ISL_3537062, EPI_ISL_3537061, EPI_ISL_3537060, EPI_ISL_3537059). It was first collected in Australia (EPI_ISL_3568416) on 2020-01-08, in Japan (EPI_ISL_2671842) on 2020-01-09, in India on 2020-01-12, and in Africa (EPI_ISL_2716636) on 2020-01-14. This variant was collected in Zhejiang (EPI_ISL_422425) and Sichuan (EPI_ISL_451345) provinces of China on 2020-01-24.

The S_614N had the third largest number with 200 genomes. It was first collected in England on 2020-03-24. The S_614S was characterized by 61 genomes, while the other genotypes had very few numbers (Table 2).

## Mutation accumulation and phylogenetic analysis of the spike variants

Similar to NS8 gene, S_614G and S_614D constituted 99.99% of the sequenced genomes (S3 Table). At the same time, S_614G was dominant over S_614D all the time. The genome number of S_614G increased exponentially from December 2019 to March 2021, and kept at high numbers (Fig 3A), while S_614D reached a peak in March 2020, and remained at low numbers for the rest of time (Fig 3B, S3 Table). Regression analysis shows that the genome number of S_614G is positively correlated with the number of confirmed cases globally with a coefficient r = 0.69 (p = 0.0031, Fig 3C), while the genome number of S_614D had no correlation (r = -0.39, p = 0.14, Fig 3D). Therefore, S_614G was the main haplotype driving the growth of the confirmed cases worldwide.

The nucleotide diversity of S_614G genomes was accumulated linearly in 2020 with $R^2$ = 0.9733 (Fig 3E), while the nucleotide diversity of S_614D increased linearly from December 2019 to August 2020, but decreased sharply in October 2020 (Fig 3F), which was probably resulted by the strict lockdown in China, since this variant mainly present in China (see below).

Phylogenetic analysis using genome sequences of S_614 alleles indicated that S_614N and S_614S were the closest relatives to the root in the ML tree (Fig 3G), however, consistent results were not obtained by using the NJ and ME methods (S4 Fig) and by the phylogenies inferred using the peptide sequences (Fig 3H, S5 Fig). Since S_614S is used in the bat spikes,

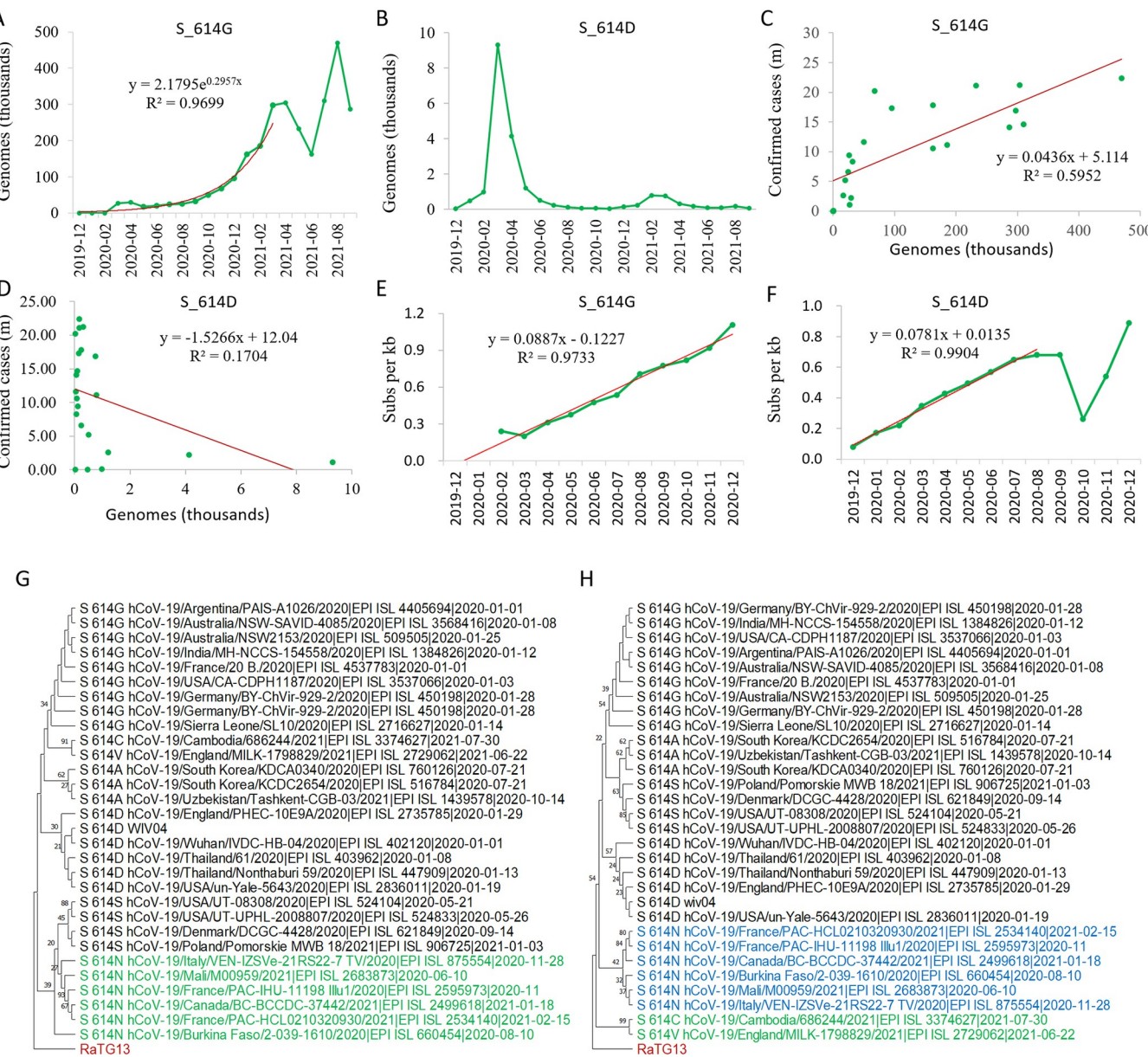

**Fig 3. Mutation accumulation and phylogenetic analysis of Spike variants in SARS-CoV-2.** A and B, Genome accumulation curve of S_614G and S_614D as counted by the sample collection date; C and D, Correlation analysis between genomes of S_614G and S_614D and confirmed cases; E and F, Mutation (Pi) accumulation curves of genotype S_614G (E) and S_614D (F); G, Maximum Likelihood tree of gene sequences of Spike as inferred by using the Tamura-Nei model [44]; H, Evolutionary tree of peptide sequences of Spike as inferred by using the Neighbor-Joining method [46]. The bootstrap supports by 1000 replicates are shown next to the branches. All positions with less than 95% site coverage were eliminated. Evolutionary analyses were conducted in MEGA11 [36].

S_614S should be the ancestral allele, and S_614N belongs to an allele that evolved early in the evolutionary history.

## Genome classification based on the two loci of NS8 and Spike

The six NS8_84 alleles and the seven spike_614 alleles should theoretically form 42 linkage types. However, only 16 haplotypes were identified in the 3.14 million genomes (Table 3).

**Table 3. Genome numbers of Spike_614 and NS8_84 mutation linkage types and their first collection date.**

| | NS8_84L | NS8_84S | NS8_84V | NS8_84I | NS8_84F | NS8_84C | total |
|---|---|---|---|---|---|---|---|
| S_614G | 2,831,567<br>2020-01-01<br>Argentina | 1,622<br>2020-02-07<br>Wuhan,China | 3<br>2020-09-15<br>South Korea | 4<br>2021-05-05<br>USA | 1<br>2021-05-06<br>Sweden | 0 | 2,833,197 |
| S_614D | 12,865<br>2019-12-24<br>Wuhan,China | 7,016<br>2019-12-30<br>Wuhan,China | 0 | 0 | 1<br>2020-02-02<br>Sichuan,China | 1<br>2020-03-12<br>USA | 19,883 |
| S_614N | 9<br>2020-03-24<br>England | 191<br>2020-06-10<br>Mali | 0 | 0 | 0 | 0 | 200 |
| S_614S | 61<br>2020-05-21<br>USA | 0 | 0 | 0 | 0 | 0 | 61 |
| S_614A | 4<br>2020-12-12<br>USA | 7<br>2020-07-13<br>South Korea | 0 | 0 | 0 | 0 | 11 |
| S_614V | 2<br>2020-07-10<br>USA | 0 | 0 | 0 | 0 | 0 | 2 |
| S_614C | 1<br>2021-07-30<br>Cambodia | 0 | 0 | 0 | 0 | 0 | 1 |
| total | 2,844,509 | 8,836 | 3 | 4 | 2 | 1 | 2,853,355 |

These haplotypes are termed by two capital letters, the first letter represents the amino acid 614 of spike, while the second letter represents the amino acid 84 of NS8. For example, S_614G +NS8_84L is termed GL, S_614G+NS8_84S is termed GS. Four main haplotypes GL, GS, DL, and DS accounted for 99.99% of the total classified genomes. The other haplotypes included GV, GI, GF, DF, DC, NL, NS, SL, AL, AS, VL, CL (Table 3).

GL haplotype was dominant over the other main haplotypes with 2.8 million sequenced genomes (99.24% of all genomes). It was dominant all the time since March 2020 with hundreds of thousands of genomes (Fig 4A). GL was first collected in Argentina (EPI_ISL_4405694) on 2020-01-01 (submission date is 2021-09-22), and was collected in USA (EPI_ISL_3537059, EPI_ISL_3537060, EPI_ISL_3537061, EPI_ISL_3537062) on 2020-01-03. This haplotype was first collected in Sichuan (EPI_ISL_451345) and Zhejiang (EPI_ISL_422425) China on 2020-01-24.

GS haplotype was first collected in Wuhan, China (EPI_ISL_412982) on 2020-02-07, which was the only sample collected globally in February 2020. It did not show up in China again, due to its rareness. Genome analysis showed that this genotype came from a recombination between GL and DS haplotype, since the S variant came from a variation of C28144T, which is closely linked with a variation of C8782T [26], however, the variation C28144T in the GS haplotype is not linked with C8782T. The GS haplotype showed up again in Spain on 2020-03-03, and a total of 19 GS genomes were identified in 2020–03 globally, including 10 in Spain, seven in USA, one in Scotland, and one in Belgium. Its number increased to a few hundred each month in 2021 (Fig 4B). All these strains did not have the linked variation C8782T, therefore, they may have all come from recombination between GL and DS haplotypes, or reverse mutations.

The DS and DL genomes reached more than a thousand only in March and April 2020 (Fig 4C), and kept at very low numbers after May 2020, possibly due to the strict lockdown policy in mainland China, since DL and DS contributed 92.6% of the genomes in mainland China before June 2020, but contributed very few proportions outside China (Fig 4D).

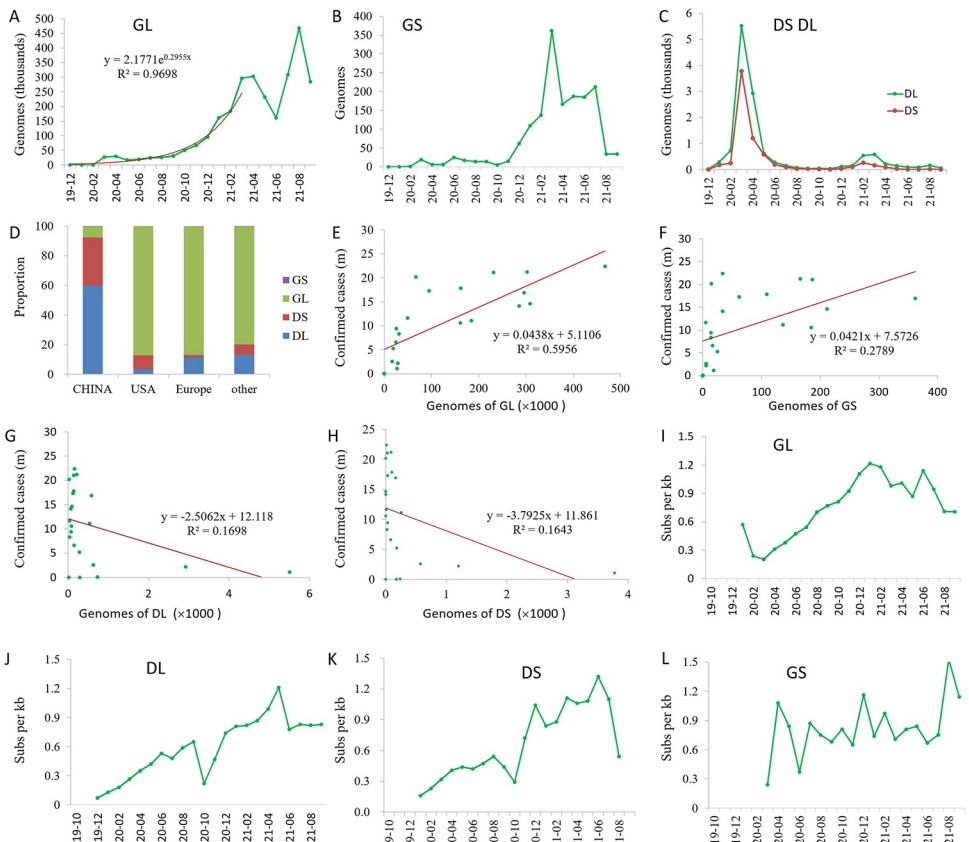

**Fig 4. Analysis of main haplotypes of SARS-CoV-2.** A-C, Genome accumulation curves of haplotype GL (A), GS (B), DL (C), and DS (C); D, Haplotype profile in China, Europe, USA, and other parts of the world; E-H, Correlation analysis between genomes of main haplotypes and confirmed cases; I-L, Mutation (Pi) accumulation curves of main genotypes.

The genome numbers and the first occurrence of other genotypes are provided in Table 3.

Regression analysis showed that GL was the major haplotype driving the increase of confirmed cases worldwide with a coefficient r = 0.77, p = 2.571e-05 (Fig 4E), while GS was weak (r = 0.53, p = 0.01152, Fig 4F). The genome numbers of DL and DS were negatively correlated with the confirmed cases, however, not significant at 5% significance level (r = -0.4, p>0.05, Fig 4G and 4H).

## Diversity dynamics of the main haplotypes

Mutation accumulation curve showed that the nucleotide diversity of haplotypes GL, DL, and DS all had a period of linear increase. The difference was that GL had a high initial Pi in January 2020 followed by a sharp decrease in February 2020 before the linear increase (Fig 4I). The nonsynonymous and synonymous mutation ratios (Ka/Ks) are between 0.65 to 0.29 (S6 Fig), indicating the presence of a strong purify selection, and many strains may have failed in further transmission till an adaptive strain emerged, which was also pointed out by Pekar and colleagues [24].

In contrast to GL haplotype, DL and DS haplotypes went straight to the linear period without initial high diversity, with correlation coefficients 0.995 (p = 3.723e-06) and 0.961 (p = 1.461e-5) for DL (Fig 4J) and DS (Fig 4K), respectively. These results suggested that DL and DS may have gone through the selection period when SARS-CoV-2 was discovered. However, the linear accumulation periods for DL and DS haplotypes were relatively shorter

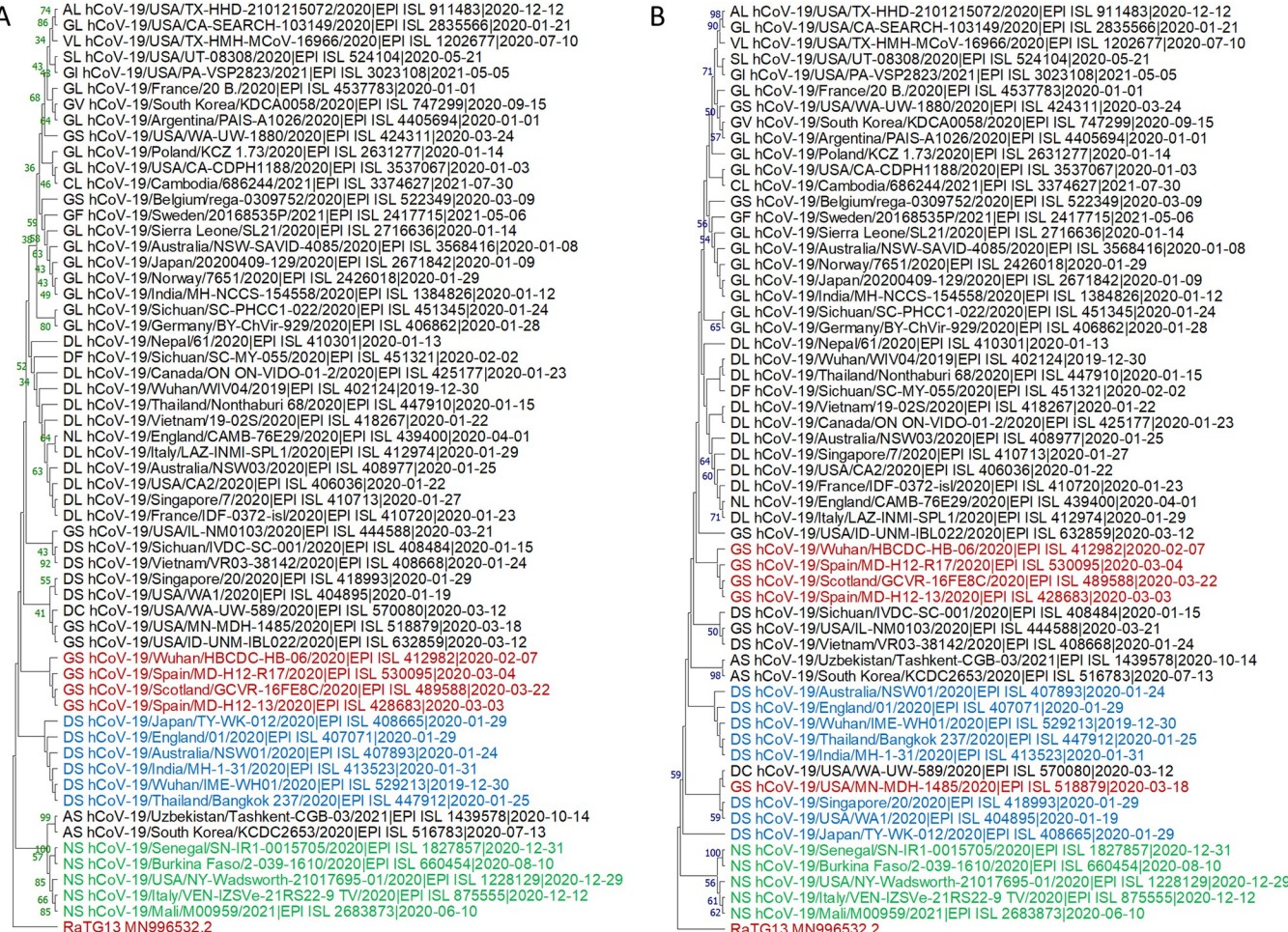

**Fig 5. Phylogenetic analysis of main haplotypes of SARS-CoV-2.** The trees were inferred by using Maximum Likelihood (A) and Neighbor-Joining (B) methods and rooted with the bat SARS-related viral genome RaTG13. For ML tree, Tamura-Nei model [44] was used, all positions with less than 95% site coverage were eliminated, and the tree with the highest log likelihood (-48997.72) is shown. For NJ tree, the evolutionary distances were computed using the Maximum Composite Likelihood method [47], and all ambiguous positions were removed for each sequence pair (pairwise deletion option). The bootstrap supports by 1000 replicates are shown next to the branches. All positions with less than 95% site coverage were eliminated. The percentages of replicate trees in which the associated taxa clustered together in the bootstrap test (1000 replicates) are shown next to the branches. Evolutionary analyses were conducted in MEGA11 [36].

compared to that of GL, possibly due to the strong transmission control measures carried out in China, since these two haplotypes were the majority haplotypes in China (Fig 4D).

The mutation accumulation curve of GS did not show any linear period, but its diversity was mostly high (Fig 4L), which was even the highest among the main haplotypes most of the months in 2020 (S7 Fig). These results suggested that GS may have evolved many times by recombination or reverse mutation, and had never obtained strong transmission ability in the human population for some reason, which was also supported by the lowest number of the sequenced genomes in the four main haplotypes (Fig 4B, S4 Table).

## Phylogenetic and phylodynamics analysis of the haplotypes

Among the three main haplotypes (GL, DL, and DS), DS is the closest haplotype to bat genomes, since most bat genomes have Spike-614D and NS8-84S [25]. The evolutionary trajectory of SARS-CoV-2 should be DS→DL→GL. The rest haplotypes are minor haplotypes

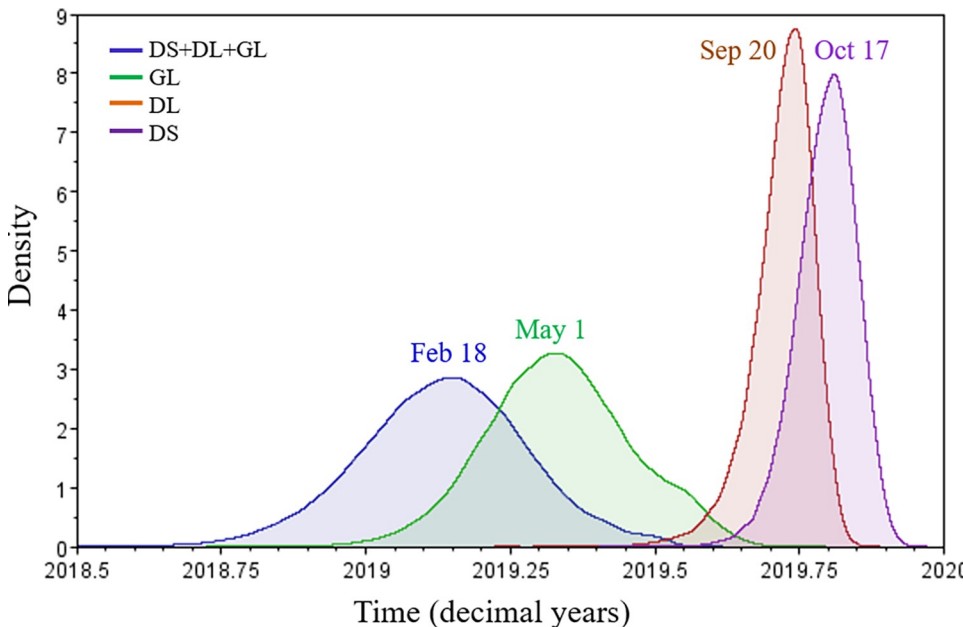

**Fig 6. Posterior distribution for the tMRCA of haplotypes DL, DS, GL, and their combination (DS+DL+GL).**
Redundant genomes were removed by using CD-HIT [38] with a threshhold of 99.97% identities. Inference was performed by using a strict molecular clock and a Bayesian Skyline coalescent prior in BEAST 1.10 [40], and summarized with Tracer 1.7 [41]. The mean tMRCA is shown above or beside the curves.

resulting from recombination, reverse mutation, and/or mutation in the adjacent sites. Phylogenetic analysis revealed that the NS haplotype together with AS and DS are the closest to the root by using all the ML, MP, NJ, and ME methods (Fig 5, S8 Fig), followed by DL and GL. Therefore, the NS, AS, and DC may have arrived from mutations of DS. The GS haplotype is distributed widely in the DS, DL, and GL haplotypes, indicating their multiple origins (recombination between GL and DS, or mutation from either DS or GL), which is in agreement with previous observation of its always high Pi (Fig 4L). NL and DF haplotypes are located in the clade of DL, indicating its origin from DL, whereas GF, GV, SL, VL, and AL were derived from GL.

Bayesian phylodynamics analysis [40] was performed to explored the evolutionary dynamics of the major haplotypes by using a Bayesian Skyline approach. Surprisingly, the newest haplotype GL in the main haplotypes was inferred to have the oldest tMRCA with a mean of May 1 (95% HPD interval: Feb 8 to Aug 4 2019, Fig 6), whereas the oldest haplotype DS had the newest tMRCA with a mean of October 17 2019 (95% HPD: September 11 to November 23 2019), while the tMRCA of DL was inferred to locate in September with a mean of September 20 (95% HPD: August 11 to October 24 2019). The tMRCA of the three haplotypes had a mean of February 18 (95% HPD: Nov 5 2018 to Jun 4 2019). These results indicate that the common ancestor of the three haplotypes may have evolved before February 2019, and the three haplotypes co-existed before April 2019. The ancestor strains of DS and DL that gave birth to GL have been extinct upon the outbreak of COVID-19, therefore, much recent tMRCAs were estimated for DS and DL by using the extant genome sequences.

## Proposed evolutionary trajectory of SARS-CoV-2

Based on the results of mutation accumulation curves, phylogenetic and phylodynamics analysis, and combined with codon usage of the mutants by single nucleotide mutation principle, we propose a possible evolutionary trajectory of SARS-CoV-2 haplotypes as shown in Fig 7.

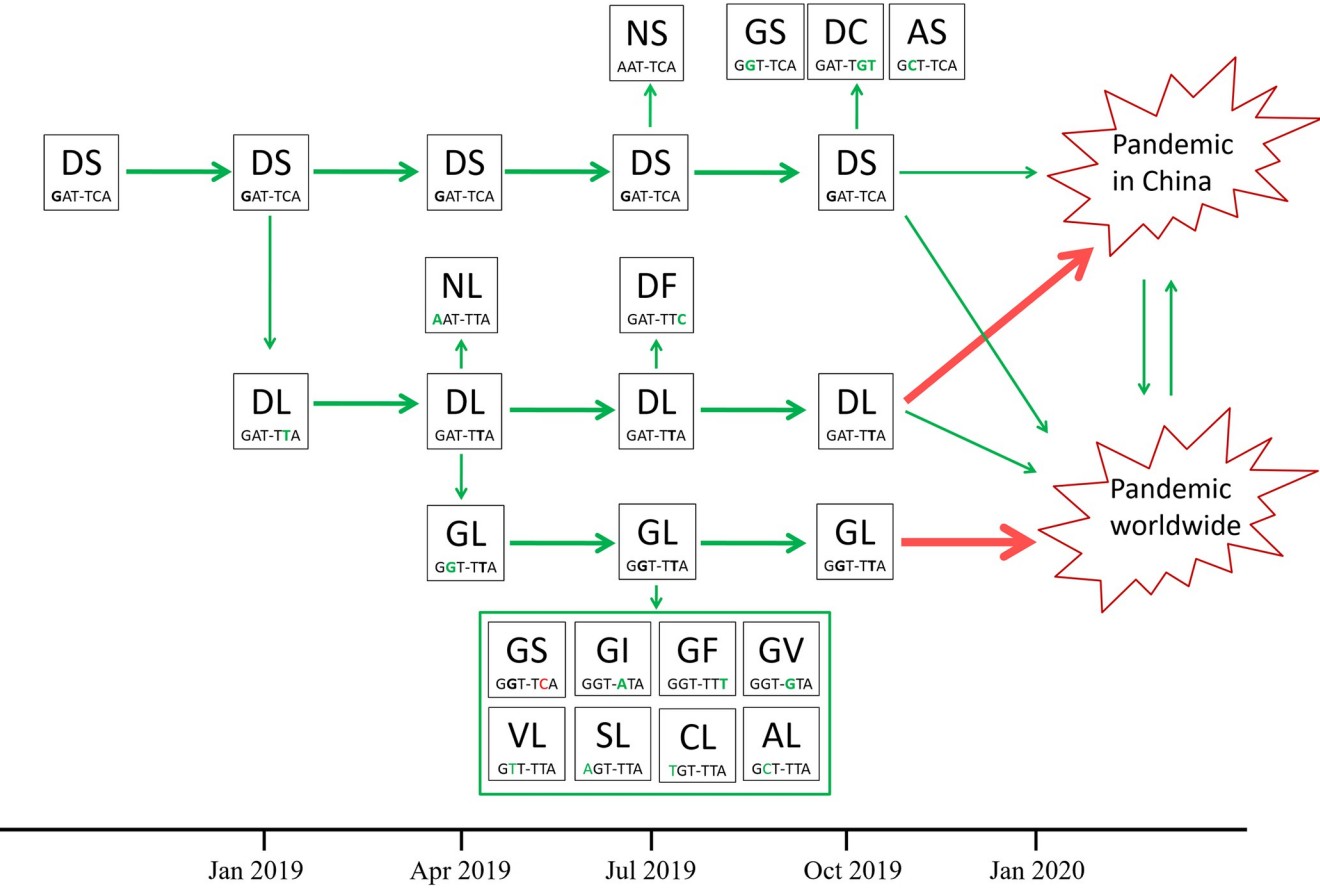

**Fig 7. Evolutionary trajectory of SARS-CoV-2 haplotypes.** The codons of S_614 and NS8_84 are presented below the haplotypes. The single nucleotides mutated from previous haplotypes were high-lighted in green. The proposed time axis is provided at the bottom. The red arrows indicate the major events.

The DS haplotypes was the ancestral haplotype of SARS-CoV-2, and may have occurred in January 2019 or earlier, and evolved to DL in February 2019, and further evolved to GL in April 2019. The ancestral strains that gave birth to GL went extinct and replaced by the more adapted newcomer at the place of its origin. However, they arrived and evolved into toxic strains in China where the GL strains had not arrived, and ignited the COVID-19 epidemic in China at the end of 2019. The GL strains had spread all over the world before the epidemic began in China, and ignited the global pandemic, which had not been noticed until it was reported in China.

## Discussion

Tracing the origin of SARS-CoV-2 is critical for preventing future spillover of the virus [48]. This is a routine process in infectious disease prevention and control. Despite tremendous efforts, the origin of SARS-CoV-2 remains unclear. Bats have been recognized as the natural reservoirs of a large variety of viruses [49]. SARS-CoV-1 that caused an epidemic in China in 2003 and the Middle East Respiratory Syndrome Coronavirus (MERS-CoV) are both suggested to be originated from bats [50–52]. A bat-origin coronavirus RaTG13 has the most similar genome compared to SARS-CoV-2 with 96.2% identities on whole genome level [53]. Several other bat-origin coronaviruses with highly similar genomic sequences compared to SARS-CoV-2 have also been found in different countries [54–56]. Although the receptor

binding domains (RBD) of the spike proteins of RaTG13 was only 89.2% compared to SARS-CoV-2, it could bind to human ACE2 (hACE2), and RaTG13 pseudovirus could transduce cells expressing hACE2 with low efficiency [57]. Conversely, the SARS-CoV-2 spike protein RBD could bind to bACE2 from *Rhinolophus macrotis* with substantially lower affinity compared with that to hACE2, and its infectivity to host cells expressing bACE2 was confirmed with pseudotyped SARS-CoV-2 virus and SARS-CoV-2 wild virus [58]. Based on these facts, SARS-CoV-2 could have more likely originated from bats. However, intermediate hosts are needed for bat-origin coronaviruses to acquire sufficient mutations so as to infect humans [59]. Two SARS-CoV-2-related coronaviruses with RBD highly similar to that of SARS-CoV-2 were found in Malayan pangolins (*Manis javanica*) [10, 60]. However, their overall genomic similarity compared to SARS-CoV-2 was both low (<93%), which suggested that pangolins were unlikely to be the intermediate host for SARS-CoV-2. So far, the direct evolutionary progenitor of SARS-CoV-2 remains unclear, whether bats are the original reservoir hosts, how, when and where SARS-CoV-2 was transmitted from animals to humans remain mysteries.

In this study, we used two gene loci to classify the haplotypes of 3.14 million SARS-CoV-2 genomes, and identified seven S_614 alleles and six NS8_84 alleles, and 16 linkage types. Phylogenetic and phylodynamics, mutation accumulation curve and codon usage analysis proposed the evolutionary trajectory of the 16 haplotypes (Fig 7). The DS haplotype was proposed to be the oldest haplotype and probably represents the haplotype of the most recent common ancestor (MRCA) of all SARS-CoV-2 strains. However, the ancestral strain of the DS haplotype was not found due to its low adaptability compared to GL haplotype. The possible occurring time of the common ancestor of the three main haplotypes was estimated by using the currently available genomes, and the mean tMRCA was estimated to be February 18 2019. Although the estimated tMRCAs of SARS-CoV-2 means the time of the most recent common ancestor of the viral variants, it should not be interpreted simply as the timing of the viral jump from animal hosts to humans, the tMRCA can be regarded as a most recent time, and the actual occurring time may be much earlier. Imagine that an ancestral SARS-CoV-2 invaded human populations, e.g. 20 years ago. Since then, the viral population may have undergone a series of strain replacements due to genetic drift and selective sweep, just like the sequential rise and fall of the delta and omicron variants. The continual losses and gains of genetic diversity may result in the re-building of the extant diversity, which may result in a very recent tMRCA. Anyway, the actual time of emergence should be much earlier than the estimated tMRCA based on the extant genome diversity.

The evolutionary trajectory we proposed suggested two major onsets of the COVID-19 pandemic (Fig 7). One was in China mainly driven by the haplotype DL, the other was driven by the haplotype GL outside China. Since GL had already spread all over the world before the virus was declared, its place of origin is not known. However, this haplotype was suggested to have emerged in Europe in a recent report by Wu and colleagues [25]. The GL haplotype had little influence in China during the early phase of the pandemic due to its late arrival and the strict transmission controls in China.

## Supporting information

**S1 Fig. Temporal curve of Theta-W (population mutation rate) of the SARS-CoV-2 globally.**
(TIF)

**S2 Fig. Molecular phylogenetic trees inferred by using peptide sequences of NS8.** A, NJ tree, B, ME tree, C, MP tree, D, ML tree. All ambiguous positions were removed for each sequence pair in NJ and ME tree, or positions with less than 95% site coverage were eliminated

in MP and ML tree. The percentage in 1000 replicates of trees in which the associated taxa clustered together is shown next to the branches. There are a total of 121 positions (NJ and ME) or 118 positions (MP and ML) in the final dataset. Evolutionary analyses were conducted in MEGA11 [36].
(TIF)

**S3 Fig. Molecular phylogenetic trees inferred by using RNA sequences of NS8.** A, NJ tree, B, ME tree, C, MP tree, D, ML tree. Codon positions included were 1st+2nd+3rd. All ambiguous positions were removed for each sequence pair in NJ and ME tree, or positions with less than 95% site coverage were eliminated in MP and ML tree. The percentage in 1000 replicates of trees in which the associated taxa clustered together is shown next to the branches. There are a total of 366 positions (NJ and ME) or 354 positions (MP and ML) in the final dataset.
(TIF)

**S4 Fig. Molecular phylogenetic trees inferred by using RNA sequences of Spike.** A, NJ tree, B, ME tree, C, MP tree, D, ML tree. Codon positions included were 1st+2nd+3rd. All ambiguous positions were removed for each sequence pair in NJ and ME tree, or positions with less than 95% site coverage were eliminated in MP and ML tree. The percentage in 1000 replicates of trees in which the associated taxa clustered together is shown next to the branches.
(TIF)

**S5 Fig. Molecular phylogenetic trees inferred by using peptide sequences of Spike.** A, NJ tree, B, ME tree, C, MP tree, D, ML tree. All ambiguous positions were removed for each sequence pair in NJ and ME tree, or positions with less than 95% site coverage were eliminated in MP and ML tree. The percentage in 1000 replicates of trees in which the associated taxa clustered together is shown next to the branches. There are a total of 121 positions (NJ and ME) or 118 positions (MP and ML) in the final dataset.
(TIF)

**S6 Fig. Accumulation curves of nonsynonymous (Ka) and synonymous (Ks) mutations.** A, Ka and Ks mutation rates; B, Ka/Ks ratios.
(TIF)

**S7 Fig. Temporal-dependent diversity (Pi) of four main genotypes.**
(TIF)

**S8 Fig. Molecular phylogenetic trees inferred by using the genome sequences.** A, ME tree, B, MP tree. All ambiguous positions were removed for each sequence pair in ME tree, or positions with less than 95% site coverage were eliminated in MP tree. The percentage in 1000 replicates of trees in which the associated taxa clustered together is shown next to the branches.
(TIF)

**S1 Table. Temporal dependent nucleotide diversity (Pi) of genomes by collection date.**
(XLSX)

**S2 Table. Mutation accumulation of main NS8_84 genotypes.**
(XLSX)

**S3 Table. Mutation accumulation of main spike genotypes.**
(XLSX)

**S4 Table. Mutation accumulation of main linkage haplotypes.**
(XLSX)

**S1 File. Metadata of SARS-CoV-2 genomes used in this study.**
(PDF)

## Acknowledgments

We gratefully acknowledge all data contributors, i.e., the Authors and their Originating laboratories responsible for obtaining the specimens, and their Submitting laboratories for generating the genetic sequence and metadata and sharing via the GISAID Initiative, on which this research is based. We are also grateful to the referees for their comments to improve the manuscript.

## Author Contributions

**Conceptualization:** Lei Yao.

**Data curation:** Xiaowen Hu.

**Formal analysis:** Jiaming Zhang.

**Funding acquisition:** Lei Yao, Jiaming Zhang.

**Investigation:** Xiaowen Hu, Yaojia Mu, Ruru Deng, Guohui Yi, Lei Yao.

**Software:** Xiaowen Hu.

**Supervision:** Jiaming Zhang.

**Writing – original draft:** Xiaowen Hu.

**Writing – review & editing:** Lei Yao, Jiaming Zhang.

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
