## [Decision Letter · Decision Letter 0]

14 Apr 2023

PONE-D-22-33104Genome charaterization based on the Spike-614 and NS8-84 loci of SARS-CoV-2 reveals two major onsets of the COVID-19 pandemicPLOS ONE

Dear Dr. Zhang,

Thank you for submitting your manuscript to PLOS ONE. After careful consideration, we feel that it has merit but does not fully meet PLOS ONE’s publication criteria as it currently stands. Therefore, we invite you to submit a revised version of the manuscript that addresses the points raised during the review process.

We look forward to receiving your revised manuscript.

Kind regards,

Babatunde Olanrewaju Motayo, Ph.D.

Academic Editor

PLOS ONE

“We are grateful to the referees for their comments to improve the manuscript. We are also grateful to those who have contributed genome sequences to the GISAID database. This research was supported by grants from National Key R&D Program of China and the Central Public-interest Scientific Institution Basal Research Fund to J.Z. (1630052020022), and the Project of Science and Technology Department of Sichuan Provincial of China to L.Y. (2019JDJQ0035).”

“This research was supported by grants from National Key R&D Program of China and the Central Public-interest Scientific Institution Basal Research Fund to J.Z. (1630052020022), and the Project of Science and Technology Department of Sichuan Provincial of China to L.Y. (2019JDJQ0035). The funders had no role in study design, data collection and analysis, decision to publish, or preparation of the manuscript.”

Reviewers' comments:

Reviewer's Responses to Questions

**Comments to the Author**

1. Is the manuscript technically sound, and do the data support the conclusions?

Reviewer #1: Yes

Reviewer #2: Yes

2. Has the statistical analysis been performed appropriately and rigorously? 

Reviewer #1: Yes

Reviewer #2: Yes

3. Have the authors made all data underlying the findings in their manuscript fully available?

Reviewer #1: No

Reviewer #2: Yes

4. Is the manuscript presented in an intelligible fashion and written in standard English?

Reviewer #1: No

Reviewer #2: Yes

5. Review Comments to the Author

Reviewer #1: The article shows important results in the area of SARS-CoV-2 and contributes to scientific knowledge.

However, there are some main concerns:

1-The number of deaths shown on page 3, line 9 must be updated. The 2021 data in the context of the pandemic is very old.

2-Although the sequences were obtained from GISAID, their deposit number is not identified according the open data principles.

3-It is suggested to cite how many and which sequences are from which geographic regions, which variant, if they are obtained from adults since the year of data collect (2021) was the year of vaccination and the vaccine can put selective pressure on viral variants.

4-In figure 1 it is not identified A and B although it is possible to assume by the graphs presented. But it is important to include the information to do it is clear to the reader.

5-On page 13, lines 12 and 13 there is a cited reference "submitted to Plos". If the data are not yet published, you can summarize the findings and refer to "unpublished data".

6-In line 18, page 13 the font size changes. Review.

7-An English review could be done.

Reviewer #2: Dear PLOS ONE Editor,

The efforts put in by the authors to deliver this highly informative report is highly commendable.

Find below my comments on the article titled: Genome charaterization based on the Spike-614 and NS8-84 loci of SARS-CoV-2 reveals two major onsets of the COVID-19 pandemic.

The article requires minor corrections of typos and other errors.

Thank you.

Minor Correction:

Title: Consider the use of the word ‘…two major possible onsets of the COVID-19 pandemic’.

Page 1, line 1: The word characterization is wrongly spelled as ‘charaterization’.

Page 3, line 7: The date (April 2019.) mentioned therein is erroneous.

Page 4, line 15: The word uncovered seems unintended in the statement ‘the origin of the virus remains uncovered’.

Page 6, line 16: The words ‘sites to be’ and ‘early’ could be expunged to avoid repetitions in the sentence.

Page 7, Table 1: The last entry designated as ‘unknown’ contain incomplete information. Kindly provide the complete information or expunge the entire cells. It may be more meaningful to fill unknowns into the cells rather than leave them blank.

Page 7, line 7: The word ‘Genome’ may not be necessary, hence the subheading may be written as ‘Mutation accumulation….’.

Page 7, Line 17: Kindly confirm that the date quoted ‘14 October 2019’ was correctly inferred from Fig 2E.

Page 7, Line 18: Kindly confirm that the date quoted ‘28 September 2019’ was correctly inferred from Fig 2F.

Page 8, line 5: The word ‘consistent’ could be expunged or corrected to the adverbial form ‘consistently’.

Page 8, lines 5 and 6: Kindly recast the phrase ‘which was supported by that the bat SARS-CoV related viruses all have NS8_84S’.

Page 8, line 16: Mutation S_614V was written twice.

Page 9, Table 2: The last entry designated as ‘unknown’ contain incomplete information. Kindly provide the complete information or expunge the entire cells. It may be more meaningful to fill unknowns into the cells rather than leave them blank.

Page 13, line 19: …went distinct. Do you mean to write extinct?

Page 13, line 27: Despite of tremendous efforts… Remove the ‘of’.

Page 13, line 29: ‘SARS-CoV-1 that caused a pandemic in China in 2003’. Ensure to correctly capture the burden and spread of the said event. A pandemic may begin in China and spread to other countries, but if the statement suggests that it affects only China, then it is an epidemic.

Figure 1: Figure 1 was not well labeled; hence it was difficult to identify which of the figures is ‘A’, ‘B’ etc.

Page 22, line 14: Figure Legend; there should be a conjunction between genomes and confirmed.

6. PLOS authors have the option to publish the peer review history of their article (what does this mean?). If published, this will include your full peer review and any attached files.

Reviewer #1: No

Reviewer #2: **Yes: **Olukunle Oluwapamilerin Oluwasemowo

---

## [Author Response · Author response to Decision Letter 0]

25 Apr 2023

Response to the academic editor and the reviewers

Dear Dr. Zhang,

Thank you for submitting your manuscript to PLOS ONE. After careful consideration, we feel that it has merit but does not fully meet PLOS ONE’s publication criteria as it currently stands. Therefore, we invite you to submit a revised version of the manuscript that addresses the points raised during the review process.

Response: Thank you for your positive comment. We have revised the manuscript according to the journal’s requirements and the points raised during review process.

Response: All these items have been included in the submission.

Response: We will try to deposit our laboratory protocols in protocols.io.

We look forward to receiving your revised manuscript.

Kind regards,

Babatunde Olanrewaju Motayo, Ph.D.

Academic Editor

PLOS ONE

Response: The manuscript has been thoroughly revised according to PLOS ONE's style requirements.

“We are grateful to the referees for their comments to improve the manuscript. We are also grateful to those who have contributed genome sequences to the GISAID database. This research was supported by grants from National Key R&D Program of China and the Central Public-interest Scientific Institution Basal Research Fund to J.Z. (1630052020022), and the Project of Science and Technology Department of Sichuan Provincial of China to L.Y. (2019JDJQ0035).”

“This research was supported by grants from National Key R&D Program of China and the Central Public-interest Scientific Institution Basal Research Fund to J.Z. (1630052020022), and the Project of Science and Technology Department of Sichuan Provincial of China to L.Y. (2019JDJQ0035). The funders had no role in study design, data collection and analysis, decision to publish, or preparation of the manuscript.”

Response: Thank you for your suggestion. The funding information has been removed from the text of the manuscript, and stated in the cover letter.

Reviewers' comments:

Reviewer's Responses to Questions

Comments to the Author

1. Is the manuscript technically sound, and do the data support the conclusions?

Reviewer #1: Yes

Reviewer #2: Yes

Response: Thank you for your positive comments.

2. Has the statistical analysis been performed appropriately and rigorously?

Reviewer #1: Yes

Reviewer #2: Yes

Response: Thank you for your positive comments.

3. Have the authors made all data underlying the findings in their manuscript fully available?

Reviewer #1: No

Reviewer #2: Yes

Response: Thank you for your comments. All the genomes used in this study are publicly available in the GISAID EpiCoV™ database (https://gisaid.org/). Supplementary figures and tables related to the research are also provided.

4. Is the manuscript presented in an intelligible fashion and written in standard English?

Reviewer #1: No

Reviewer #2: Yes

Response: Many thanks for your corrections and suggestions on improving the quality of the manuscript.

5. Review Comments to the Author

Reviewer #1: 

The article shows important results in the area of SARS-CoV-2 and contributes to scientific knowledge.

Response: Many thanks for your positive comments.

However, there are some main concerns:

1-The number of deaths shown on page 3, line 9 must be updated. The 2021 data in the context of the pandemic is very old.

Response: Thank you for your suggestion. The data has been updated through to 23 April 2023.

2-Although the sequences were obtained from GISAID, their deposit number is not identified according the open data principles.

Response: Thank you for your suggestion. The data availability through the associated EPI_SET Identifier for all genome sequences and metadata used in the analysis was provided in S1 File, through which a link to the metadata and genomes in the GISAID database is provided. From the link, one can view the contributors of each individual sequence with details such as accession number, Virus name, Collection date, Originating Lab and Submitting Lab and the list of Authors.

3-It is suggested to cite how many and which sequences are from which geographic regions, which variant, if they are obtained from adults since the year of data collect (2021) was the year of vaccination and the vaccine can put selective pressure on viral variants.

Response: Thank you for your suggestion. Our study was based on the genomes submitted till 2021-10-18, and the sample collection dates range from 2019-12-24 to 2021-10-16. Main studies in this manuscript selected the genomes in the early phase of the pandemic when vaccine had not been applied and the diversity of the viral genomes increased linearly. Vaccine may have an influence on diversity and may contributed to the rise and fall of diversity in the late phase of the pandemic. We have discussed about this issue in the manuscript. We have prepared a GISAID Identifier: EPI_SET_230423ot in S1 File, through the link, one can view detailed information, including geographic regions, variants, Collection date, Originating Lab and Submitting Lab and the list of Authors.

4-In figure 1 it is not identified A and B although it is possible to assume by the graphs presented. But it is important to include the information to do it is clear to the reader.

Response: Thank you for your correction. Identifiers A, B, C, and D have been included in the revised figure.

5-On page 13, lines 12 and 13 there is a cited reference "submitted to Plos". If the data are not yet published, you can summarize the findings and refer to "unpublished data".

Response: Thank you for your suggestion. The manuscript cited in the text is still under review. The findings in that manuscript has been summarized into one sentence and referred to as "unpublished data".

6-In line 18, page 13 the font size changes. Review.

Response: Thank you for your correction. The font size of all text in the manuscript has been changed according to the guidance of Plos One.

7-An English review could be done.

Response: Thank you for your suggestion. The text has been reviewed by an English expert.

Reviewer #2: 

Dear PLOS ONE Editor,

The efforts put in by the authors to deliver this highly informative report is highly commendable.

Find below my comments on the article titled: Genome charaterization based on the Spike-614 and NS8-84 loci of SARS-CoV-2 reveals two major onsets of the COVID-19 pandemic.

The article requires minor corrections of typos and other errors.

Thank you.

Response: Thank you for your positive comments on our manuscript.

Minor Correction:

Title: Consider the use of the word ‘…two major possible onsets of the COVID-19 pandemic’.

Response: Thank you for your suggestion. We have changed accordingly.

Page 1, line 1: The word characterization is wrongly spelled as ‘charaterization’.

Response: Thank you for your correction.

Page 3, line 7: The date (April 2019.) mentioned therein is erroneous.

Response: Thank you for your correction. The date should be April 2020.

Page 4, line 15: The word uncovered seems unintended in the statement ‘the origin of the virus remains uncovered’.

Response: Many thanks for your correction. The word has been revised.

Page 6, line 16: The words ‘sites to be’ and ‘early’ could be expunged to avoid repetitions in the sentence.

Response: Thank you for the correction. The sentence has been thoroughly revised to be precise.

Page 7, Table 1: The last entry designated as ‘unknown’ contain incomplete information. Kindly provide the complete information or expunge the entire cells. It may be more meaningful to fill unknowns into the cells rather than leave them blank.

Response: Thank you for your suggestion. We have filled the blank cells.

Page 7, line 7: The word ‘Genome’ may not be necessary, hence the subheading may be written as ‘Mutation accumulation….’.

Response: Thank you for your suggestion. The subheading has been revised accordingly.

Page 7, Line 17: Kindly confirm that the date quoted ‘14 October 2019’ was correctly inferred from Fig 2E.

Response: Thank you for the remind. Fig 4E was drawn using Excel, it is just a sketch map showing the trend, and is not accurate. For accurate estimation, the dates and the 95% confidence intervals were inferred by using regression analysis, but not directly from the figure. It was the citing position that caused the misunderstanding. The citing has been put in the correct place in the revised manuscript.

Page 7, Line 18: Kindly confirm that the date quoted ‘28 September 2019’ was correctly inferred from Fig 2F.

Response: Many thanks for the remind. This question is similar to the one above. The dates were estimated by using regression analysis, not inferred directly from the figure. The figure is just a sketch map and not accurate enough to inferred dates.

Page 8, line 5: The word ‘consistent’ could be expunged or corrected to the adverbial form ‘consistently’.

Response: Thank you for your correction. The word ‘consistent’ has been deleted.

Page 8, lines 5 and 6: Kindly recast the phrase ‘which was supported by that the bat SARS-CoV related viruses all have NS8_84S’.

Response: Many thanks for your suggestion. The sentence has been revised.

Page 8, line 16: Mutation S_614V was written twice.

Response: Thank you for your correction.

Page 9, Table 2: The last entry designated as ‘unknown’ contain incomplete information. Kindly provide the complete information or expunge the entire cells. It may be more meaningful to fill unknowns into the cells rather than leave them blank.

Response: Thank you for your suggestion. The last entry designated as ‘unknown’ are genomes that could not be classified due to incomplete sequencing or low quality sequences in allele. The entire cells were expunged since they did not provide useful information.

Page 13, line 19: …went distinct. Do you mean to write extinct?

Response: Thanks for your correction. I mean to write extinct. The word has been corrected.

Page 13, line 27: Despite of tremendous efforts… Remove the ‘of’.

Response: Thank you for your correction. It has been removed.

Page 13, line 29: ‘SARS-CoV-1 that caused a pandemic in China in 2003’. Ensure to correctly capture the burden and spread of the said event. A pandemic may begin in China and spread to other countries, but if the statement suggests that it affects only China, then it is an epidemic.

Response: Thank you for your suggestion. The word has been changed.

Figure 1: Figure 1 was not well labeled; hence it was difficult to identify which of the figures is ‘A’, ‘B’ etc.

Response: Thank you for your correction. The labels have been added.

Page 22, line 14: Figure Legend; there should be a conjunction between genomes and confirmed cases.

Response: Yes, you are right. The sequenced-genome number and the confirmed cases are positively correlated with a coefficient r=0.75 (p-value = 5.5e-05), which means that the more confirmed cases, the more virus genomes were sequenced.

6. PLOS authors have the option to publish the peer review history of their article (what does this mean?). If published, this will include your full peer review and any attached files.

Do you want your identity to be public for this peer review? For information about this choice, including consent withdrawal, please see our Privacy Policy.

Reviewer #1: No

Reviewer #2: Yes: Olukunle Oluwapamilerin Oluwasemowo

Response: Thank you for your kind review.

Response: The figures have been treated by PACE.

---

## [Decision Letter · Decision Letter 1]

4 Jun 2023

Genome characterization based on the Spike-614 and NS8-84 loci of SARS-CoV-2 reveals two major possible onsets of the COVID-19 pandemic

PONE-D-22-33104R1

Dear Dr. Zhang,

We’re pleased to inform you that your manuscript has been judged scientifically suitable for publication and will be formally accepted for publication once it meets all outstanding technical requirements.

Kind regards,

Babatunde Olanrewaju Motayo, Ph.D.

Academic Editor

PLOS ONE

Additional Editor Comments (optional):

Reviewers' comments:

Reviewer's Responses to Questions

**Comments to the Author**

1. If the authors have adequately addressed your comments raised in a previous round of review and you feel that this manuscript is now acceptable for publication, you may indicate that here to bypass the “Comments to the Author” section, enter your conflict of interest statement in the “Confidential to Editor” section, and submit your "Accept" recommendation.

Reviewer #2: All comments have been addressed

2. Is the manuscript technically sound, and do the data support the conclusions?

Reviewer #2: Yes

3. Has the statistical analysis been performed appropriately and rigorously? 

Reviewer #2: Yes

4. Have the authors made all data underlying the findings in their manuscript fully available?

Reviewer #2: Yes

5. Is the manuscript presented in an intelligible fashion and written in standard English?

Reviewer #2: Yes

6. Review Comments to the Author

Reviewer #2: Dear Editor,

I am pleased to see that all the concerns previously raised have been addressed by the author(s).

However, I will expect to see in the final production phylogenetic trees with clearer taxon names and fonts rather that the blurry images.

Thank you.

7. PLOS authors have the option to publish the peer review history of their article (what does this mean?). If published, this will include your full peer review and any attached files.

Reviewer #2: No

---

## [Editor Report · Acceptance letter]

8 Jun 2023

PONE-D-22-33104R1 

Genome characterization based on the Spike-614 and NS8-84 loci of SARS-CoV-2 reveals two major possible onsets of the COVID-19 pandemic 

Dear Dr. Zhang:

I'm pleased to inform you that your manuscript has been deemed suitable for publication in PLOS ONE. Congratulations! Your manuscript is now with our production department. 

Kind regards, 

on behalf of

Dr Babatunde Olanrewaju Motayo 

Academic Editor

PLOS ONE